# Deepening Depression in Women Balancing Work–Life Responsibilities and Caregiving during the COVID-19 Pandemic: Findings from Gender-Specific Face-to-Face Street Interviews Conducted in Italy

**DOI:** 10.3390/bs13110892

**Published:** 2023-10-29

**Authors:** Laura Giusti, Silvia Mammarella, Sasha Del Vecchio, Anna Salza, Massimo Casacchia, Rita Roncone

**Affiliations:** 1Department of Life, Health and Environmental Sciences, University of L’Aquila, 67100 L’Aquila, Italy; laura.giusti@univaq.it (L.G.); silvia.mammarella@univaq.it (S.M.); sasha.delvecchio@graduate.univaq.it (S.D.V.); anna.salza@univaq.it (A.S.); massimo.casacchia@univaq.it (M.C.); 2University Unit for Rehabilitation Treatment, Early Interventions in Mental Health, S. Salvatore Hospital, 67100 L’Aquila, Italy

**Keywords:** women, depression, family burden, family functioning, COVID-19 pandemic, collective traumas

## Abstract

Purpose: This study investigated the impact of the COVID-19 pandemic on mental health, quality of life, and family functioning in a sample of the general female population, exploring difficulties encountered in managing family and work responsibilities and burden of care when taking care of a loved one. This study was, moreover, aimed at investigating factors capable of influencing severe depressive symptomatology in the context of socio-demographics, traumatic events, individual vulnerability, and family functioning. Method: The sampling method used in this research was non-probability sampling. The survey took place during a Hospital Open Weekend (8–10 October 2021) organized by the National Gender Observatory on Women’s Health “Fondazione Onda” on the occasion of the World Mental Health Day. Results: A total of 211 women were interviewed (mean age = 35.6, 53% living alone, more than 15% with financial difficulties, 47% exposed to the 2009 L’Aquila earthquake). More than 50% of the sample reported a higher complexity in managing their lives during the COVID-19 pandemic compared to their previous routine, with no statistically significant differences between working women and non-workers, although the latter obtained higher scores for depressive symptomatology and poorer quality of life. Compared to non-caregivers, female caregivers (22.3%) in charge of the care of loved ones affected by physical (10.9%) or psychiatric disabilities (11.4%) complained of a poorer quality of life, especially in general health perception (*p* = 0.002), physical function (*p* = 0.011), role limitations related to physical problems (*p* = 0.017), bodily pain (*p* = 0.015), mental health (*p* = 0.004), and social functioning (*p* = 0.007). Women caring for people affected by mental disorders seemed to experience a more significant worsening in vitality (*p* = 0.003) and social functioning (*p* = 0.005). Approximately 20% of the total sample reported severe depressive symptomatology. Previous access to mental health services (O.R. 10.923; *p* = 0.000), a low level of education (O.R. 5.410; *p* = 0.021), and difficulties in management of everyday lives during the COVID-19 pandemic (O.R. 3.598; *p* = 0.045) were found to be the main variables predictive of severe depressive psychopathology. Old age, good problem-solving skills, and ability to pursue personal goals were identified as protective factors. Conclusions: The COVID-19 pandemic underlined the need for support amongst emotionally vulnerable women with pre-existing mental health conditions, partly reflecting the cumulative effects of traumas.

## 1. Introduction

During the early waves, distancing and reduced social contacts proved to be the most effective means of slowing down the COVID-19 pandemic. However, measures applied to contain the spread of COVID-19 dramatically changed how people worked, lived, and studied, with numerous organizations switching to remote working and many planning to continue this practice in the long term. Changes in the non-work domain have been equally dramatic; new responsibilities have emerged for many, while interpersonal resources have shrunk [1,2].

Numerous international studies have highlighted that the COVID-19 pandemic has led to high levels of psychological distress in the general population [3,4,5,6,7] and in the educational and academic fields [8,9,10,11,12]. Studies have also confirmed how social distancing, implemented in response to the first wave of COVID-19, coincided with a worsening of mental health across multiple countries [13,14,15]. The feeling of loneliness and conditions of uncertainty constitute key risk factors for short- and long-term consequences on mental health and affect how we interact with others [16,17,18]. Loneliness can affect physiological and mental health [19]. During the pandemic, seeing others as a potential threat of infection and the need to isolate oneself can further lead to maladaptive strategies, such as increased avoidance behavior. As a result, lonely individuals perceive social interactions more negatively, which is reinforced by pandemic containment measures [19,20].

Several studies conducted in Italy have investigated the negative psychological consequences produced by the pandemic on the general population, taking into account predictors such as female gender, infection of an acquaintance and/or a family member, history of medical problems, having been subjected to stressful and traumatic situations such as previous natural disasters [21,22], and availability of inadequate physical space during isolation [10,23,24]. Moreover, the severe impact on the population’s mental health coincided with a drastic reduction in levels of care [25].

With regard to the effect of COVID-19 on mental health, a large number of studies have highlighted that in the female sex, there is a significant association between higher self-reported levels of stress, anxiety, depression, and post-traumatic stress symptoms and a more severe overall psychological impact [4,26,27,28]. Compared to men, the increased prevalence of depressive disorders observed in women underlined the presence of a highly significant difference produced as a result of the social and economic consequences of the pandemic [29]. Indeed, females are characterized by a higher prevalence of risk factors known to intensify during a pandemic, including chronic environmental strain [30], pre-existing depressive and anxiety disorders [31], and domestic violence [32,33,34]. 

Furthermore, COVID-19 seems to have amplified gender inequalities in the work domain [35,36]. Throughout the COVID-19 pandemic and associated compulsory lockdowns, working women undoubtedly found it challenging to cope with changes in the workplace and adapt to remote working routines [37,38,39]. Mothers who were suddenly expected to balance remote working with family life were affected by a series of physical, mental, and social issues, including anxiety, stress, sleep deprivation, and relationship problems [36]. The results of a cross-sectional survey on European working women from five countries, including France, Italy, Poland, Sweden, and the United Kingdom, revealed how women working from home displayed a higher prevalence of depressive symptoms compared to those who commuted to work, suggesting that networking with people face-to-face acted as a significant protective factor against experience of symptoms of depression during a period of social distancing [40].

In addition to having to work from home, women were often required to shoulder the burden of additional caring responsibilities, such as supporting children during online education or taking care of a family member with a physical or mental disability. Women assumed the role of caregiver more often than men, scored lower on a quality-of-life measure, and reported higher levels of anxiety [41,42,43]. Caregivers reported a limited ability to cope with life stressors and increased social isolation [44]. One year into the pandemic, female caregivers of dementia patients tended to display more stress-related symptoms compared to baseline, including depression, anxiety, irritability, and anguish [45]. Caregivers of people affected by schizophrenia reported a heavy burden of care and high levels of stress during the pandemic [46,47]. Indeed, even prior to the onset of the COVID-19 pandemic, the quality of life amongst caregivers of schizophrenic patients had been particularly poor [48]. Throughout the COVID-19 pandemic, caregivers’ concerns were focused not only on their own health but also on the continuity of care and well-being of their family members who were affected by mental illnesses [49,50] and disabilities [51]. 

In Italy, the ONDA Foundation, a National Gender Observatory on Women’s Health established in Milan, has been working since 2005 to promote women’s health by carrying out a series of activities and projects (gender surveys, events, publications, digital campaigns, and thematic focuses). The ONDA Foundation collaborates with the National Health Service and social health structures. Every year, the Foundation recognizes the focus of Italian hospitals on gender-specific medicine by conferring the “Bollini rosa” award. With the support of a virtual network of 354 hospitals in Italy, the ONDA Foundation organizes Hospital (H) Open Days, (H) Open Weeks, and (H) Open Weekends, which are free initiatives aimed at facilitating diagnosis and access to appropriate treatment in the female population. One of the targets of these activities is to raise awareness of mental health issues, particularly depression, in the female population.

In the context of the initiatives carried out by the ONDA network, our study aimed to (1) evaluate depressive symptomatology, quality of life, and family functioning, with a particular focus on the difficulties faced by women during the COVID-19 pandemic, in a general population sample by comparing working and non-working women; (2) compare women caring for a loved one affected by physical or psychiatric disabilities and non-caregivers; and (3) identify factors influencing the onset of severe depressive symptomatology in the context of socio-demographics, traumatic events, and individual vulnerability and family functioning skills.

Using the conceptual framework of a psychosocial vulnerability model [52] including socio-demographic (age, educational level, and civil status), clinical (pre-existing psychological problems), contextual (previous traumatic events and family functioning), and psychosocial (caregiver role) variables, we were interested in evaluating the factors impacting higher depressive symptomatology amongst women during the COVID-19 pandemic.

We hypothesized that (1) working women could be more distressed and be more vulnerable to experience persistent feeling of sadness due to household and life management during the COVID-19 pandemic, and (2) women taking care of a loved one or with low family functioning might be at increased risk of developing severe depressive symptoms.

## 2. Methods

### 2.1. Study Design and Population

The survey took place during a Hospital Open Weekend (8–10 October 2021) organized by the National Gender Observatory on Women’s Health “Fondazione Onda” on the occasion of the World Mental Health Day.

The University Unit for Rehabilitation Treatment and Early Interventions in Mental Health, known as TRIP, located within the S. Salvatore Hospital and directed by Prof. Rita Roncone, took part in the initiative to raise awareness amongst women of the importance of defending their mental well-being and the right to “Re-start” from their own life goals, while encouraging them, in case of distress, to seek early diagnosis and access to treatment, and helping them overcome fears, prejudice, and stigma related to mental disorders.

The team included psychiatrists, researchers, and Ph.D. students in clinical psychology, as well as undergraduate students specializing in psychiatric rehabilitation techniques, who conducted the questionnaire-based ‘face-to-face’ street interviews and collected data online. The team attended a short 4 h training session on interview techniques and early identification of emotional distress.

The entire team set up a station in the Centre of L’Aquila where women were able to voice their concerns and emotions and where interviewers could suggest strategies to help women improve their quality of life.

The sampling method used in this research was non-probability sampling. Street interviews are one of the fastest and most accurate forms of real-time data collection. In this form of data collection, women who had been approached on the street and agreed to participate, were asked targeted questions included in the survey “I start from myself” in the form of a structured interview. At the end of the interview, they were all encouraged to access the service for a free comprehensive psychiatric consultation, if they thought it might be helpful.

The interview questions were derived from the results of an online focus group that took place through Microsoft 365 ^®^ Teams (Microsoft Corporation, Redmond, WA, USA), which was set up to develop concepts and questions for the interview design. The focus group meeting lasted two hours and included all the authors of this study, who had all been exposed to the 2009 L’Aquila earthquake. Two female users of the TRIP Service who had a history of depressive disorder and were facing burden of care as caregivers of family members affected, respectively, by physical or mental disabilities participated in the focus group.

### 2.2. Context

In April 2009, an earthquake with a magnitude of 6.3 hit the province of L’Aquila, claiming the lives of 309 people, injuring thousands of citizens, causing tens of thousands of displaced people, and provoking severe material destruction [53,54,55]. Reconstruction of the city is still ongoing, as is the process of “remediation” of the psychosocial vulnerability of the exposed population [56,57,58]. In this study, we also focused on the variables related to exposure to and impact produced by experiencing the April 2009 earthquake, which was considered a crucial collective life event for people living in the area.

### 2.3. Assessment Tools

The assessment consisted of three parts:(a)Section 1 included information on the study, privacy protection, and informed consent.(b)Section 2 included the participants’ demographic backgrounds, including age, education, work, marital status, number of children, working activity, and socio-economic status. The history of life events included the impact of the COVID-19 pandemic, complex management of family life and work during the COVID-19 pandemic, and the impact produced by the 2009 L’Aquila earthquake measured on a 5-point Likert scale (0 = None; 1 = Only a little; 2 = To some extent; 3 = Considerably; 4 = Greatly). Previous contact with mental health services, mental health issues, and treatments were also assessed.(c)Section 3 included standardized questionnaires investigating quality of life, psychopathology, family functioning, and family burden.

The Patient Health Questionnaire (PHQ-9) [59] was a tool applied in this study to evaluate depressive symptoms and levels of severity. It contains nine items rated on a four-point Likert scale (0 = not at all; 3 = nearly every day). The PHQ-9 total score for the nine items ranges from 0 to 27. A PHQ-9 score ≥ 10 has a sensitivity of 88% and a specificity of 88% for major depression. PHQ-9 scores of 5, 10, 15, and 20 represent mild, moderate, moderately severe, and severe depression, respectively [60]. For the purpose of this study, we used a cut-off score of 10. The internal reliability was excellent, with a Cronbach’s alpha of 0.89 (Kroenke et al., 2001). Our sample’s internal consistency for the PHQ-9 was high (Cronbach’s α = 0.87).

The 36-Item Short Form Survey (SF-36) [61,62] is a self-reported measure of a population’s health-related quality of life (QoL). The SF-36 is a 36-item form that measures eight different dimensions of health: general health perception (GH), physical function (PF), role limitations related to physical problems (RF), bodily pain (BP), mental health (MH), role limitations due to emotional problems (RE), vitality (VT), and social functioning (SF). Raw scores are linearly transformed into 0–100 scales. Higher transformed scores indicate better health.

Family functioning was assessed using the Family Functioning Questionnaire (FFQ) [63], which was developed to assess family functioning pattern under the framework of psychoeducational family interventions. This questionnaire consists of 24 items. It measures the following three dimensions:

(1) Problem solving (eight items) refers to the six steps of structured problem solving by identifying the problem or the objective, listing possible alternative solutions, discussing the positive and negative aspects of each proposal, choosing the best solution (or a better and more satisfying and realistic solution), planning the solution, and checking and reviewing the implementation and planning. (2) Communication skills (eight items) are concerned with the expression of positive and negative feelings, the making of requests, and active listening skills (probing questions and providing a summary of what has been understood). (3) Personal goals (eight items) are defined as the ability of each family member to identify everyday personal goals (not linked to subject care). Responses range from 1 “never” to 4 “always”. Higher scores are indicative of healthier functioning.

Items are evaluated on a 4-point Likert scale; a higher score is associated with better family functioning (range 24–96). The scale was originally developed and standardized in the Italian population and has demonstrated good internal consistency (Cronbach’s alpha coefficient ranges from 0.75 to 0.84 for the three dimensions) and test–retest reliability (Pearson’s r correlation coefficient ranges from 0.75 to 0.60) [63]. The internal consistency for the FFQ in our sample was high (Cronbach’s a = 0.88). 

The version of the Family Problem Questionnaire, FPQ [64] used in this study consisted of a shortened version of the 44-item instrument [65], which had recently been utilized in an Italian multicentric family study [66]. In this study, we selected sections specifically aimed at assessing the objective and subjective burden of care and the dimension of support received (from professionals, relatives, and friends). We investigated the following: (1) objective burden (twelve items, range 13–52) related to the impact on daily activities/social life; (2) subjective burden (six items, range 6–24) related to the impact on caregiver well-being, distress over the condition of the affected family member, and concern for the future; (3a) professional support received (four items, range 4–16); and (3b) support from relatives and friends (three items, range 3–12).

Items are evaluated on a 4-point Likert scale. Higher scores are associated with a higher burden of care and scarce support from professionals, relatives, and friends.

### 2.4. Statistical Analyses

Statistical analyses were conducted in four phases: (1) a descriptive analysis of socio-demographics, clinical data, health-related quality of life, depression, family functioning, and burden of care in female caregivers was performed; continuous variables were reported as means (standard deviations), and categorical variables were reported as frequencies (percentages). (2) Baseline comparisons [chi-square, *t*-tests, and one-way analysis of variance (ANOVA)] were performed to assess differences between female caregivers and non-caregivers and between non-caregivers and caregivers of family members affected by either physical or mental disabilities. Bonferroni post hoc correction was calculated. (3) A correlation analysis (r, Pearson correlation) was conducted to verify the relationships between caregivers’ age and years of education and five out of the eight dimensions of health-related quality of life as measured using the SF-36, the four-subscales of burden of care as measured using the FPQ, and the three-dimensions of family functioning (FFQ). (4) Multinominal logistic regression analyses were conducted to identify variables capable of influencing depressive symptomatology. The dependent variable, depression (based on PHQ-9 scores), was coded as 1 = absent–mild depression (PHQ-9 scores 0–5); 2 = moderate depression (PHQ-9 scores 6–10); 3 = moderately severe depression (PHQ-9 scores 11–15); and 4 = severe depression (PHQ-9 scores >15).

The independent variables in the model included women’s age; not having a stable affective partnership; low educational level; financial difficulties; having traumatically experienced the 2009 L’Aquila earthquake; having contracted COVID-19; complex life management during the COVID-19 pandemic; previous access to mental health services; caregiving for a loved one; and the three dimensions of the FFQ (problem solving, communication, and personal goals).

Not having a stable affective partnership was coded into two categories (1 = single, separated/divorced, or widowed; 0 = married). Education was coded into two categories (1 = less than 13 years of education; 0 = 13 years or more of education, i.e., graduation or higher). Economic difficulties, having traumatically experienced the 2009 L’Aquila earthquake, having contracted COVID-19, complex life and household management during the COVID-19 pandemic, previous contact with mental health services, and caregiving for a loved one were coded into two categories (1 = yes; 0 = no).

With regard to our model, the selection of the independent variables was based mostly on previous research. “Age”, as an independent variable, was included to estimate the ability of younger women to better manage disasters [67] and distress, particularly amongst psychiatric caregivers [68,69]. The inclusion of the independent variables related to collective traumatic events and their consequences (severe long-term impact of the 6 April 2009 L’Aquila earthquake; having contracted COVID-19 virus infection; complex management of family life and work during the COVID-19 pandemic) was motivated by the vulnerability of women to collective traumatic events such as earthquakes [21,22,70], and the recent collective trauma of the COVID-19 pandemic [71,72,73]. Amongst women, isolation, economic precarity, and previous mental health issues were predictors of traumatic conditions during COVID-19 [71,74] in view of the complexity of managing the household and their own lives during the COVID-19 pandemic, including having to continuously juggle the work–family balance [36,75]. In our model, the three dimensions of family functioning were selected based on the assumption that low family functioning could be predictive of depression [76,77,78].

Statistical analyses were conducted using SPSS 27.0 (SPSS Inc., Chicago, IL, USA). All tests were two-tailed, and *p* < 0.05 was considered significant.

## 3. Results

### 3.1. Socio-Demographic and Characteristics of the Sample, Depression, Health-Related Quality of Life, and Family Functioning

Table 1 describes the main socio-demographic and clinical characteristics of the 211 women who took part in the study.

The mean age of the total sample was 35.6 (SD = 18.5) (range: 18–82). The majority of the women were Italian, with less than 5% (*n* = 10) originating from Moldova, Ukraine, Albania, Iran, and Argentina. More than half of the sample (53.1%, *n* = 112) comprised women living alone without a stable affective relationship.

Approximately 70% of this sample of young women (43% students) had no children, and less than 35% were employed in a paid position, whilst more than 40% held a university degree. Slightly more than 15% complained of financial difficulties.

#### 3.1.1. Comparison between Female Workers and Non-Workers

Statistically significant differences were detected between the two groups (workers and non-workers). Non-working women were characterized by a younger age than working women (*t*-test: −3.598; *p* = 0.000), were more likely to be married (chi-square: 27.970; f.d. 3; *p* = 0.000) and have children (chi-square: 10.858; f.d. 3; *p* = 0.001), and possessed a lower level of education (chi-square: 26.201; f.d. 3; *p* = 0.000) (Table 1).

More than 50% of the sample (N = 108, 51.2%) reported complexities in managing their lives during the COVID-19 pandemic compared to pre-pandemic times (Table 2); no statistically significant differences were detected between workers (42.5%) and non-workers (55.8%). Ten percent of the sample had contracted COVID-19 infection, and 13.3% reported having lost loved ones due to COVID-19.

Only a small proportion of women (7.6%; *n* = 16) had refused COVID-19 vaccination. These decisions were not found to be related in a statistically significant manner to socio-demographics, level of education (some women held a university degree), or clinical variables. 

Less than half of the sample (*n* = 100, 47.4%) were exposed to the 2009 L’Aquila earthquake; among these women, 43% reported a severe impairment in at least two of the three dimensions investigated (family life, work, or social life), while 7.1% (*n* = 15) confirmed having lost someone close. Compared to non-workers, a higher proportion of female workers were exposed to the 2009 L’Aquila earthquake (chi-square: 9.091; f.d. 1; *p* = 0.003), with the catastrophic event producing a severe impact on their lives (chi-square: 6.550; f.d. 1; *p* = 0.010).

Almost 45% of the total sample (*n* = 94) reported having previously accessed mental health services, with no statistically significant differences between those worked and those who did not (Table 2). Psychopharmacological treatment was prescribed to 18.5% of the total sample, with no statistically significant differences between the two groups with regard to subsequent integrated therapies (psychopharmacological plus psychotherapeutic treatments) or psychotherapeutic therapies alone.

Forty percent of the sample obtained a PHQ-9 score higher than the cut-off score of 10, with approximately 20% being diagnosed as being affected by severe depression based on their PHQ-9 scores (Table 3). A statistically significant difference was found in terms of PHQ-9 scores between the two groups of women, with non-workers obtaining higher scores (*t*-test: 2.936; *p* = 0.004). A higher proportion of non-workers (*n* = 63, 76.8%) had a PHQ-9 score exceeding the cut-off score of 10 compared to working women (58.9%) (chi-square = 7.157; f.d. = 1; *p* = 0.005). 

Statistically significant differences were found between the two groups of women in family functioning dimensions, with working women showing higher scores in communication skills (*t*-test: −3.496; *p* = 0.001) and problem solving (*t*-test: −4.118; *p* = 0.000) and lower scores in pursuing personal goals (*t*-test: 4.027; *p* = 0.000) compared to non-workers (Table 3).

Compared to workers, non-workers obtained lower scores in all quality-of-life domains related to mental health, MH (*t*-test: −3.148; *p* = 0.002), RE (*t*-test: −3.585; *p* = 0.000), VT (*t*-test: −3.090; *p* = 0.002), and SF (*t*-test: −2.843; *p* = 0.005) (Figure 1).

No statistically significant differences were revealed in the proportion of workers (*n* = 32) and non-workers (*n* = 15) who acted as caregivers (*n* = 47, 22.3%) or were taking care of loved ones with physical (*n* = 23, 10.9%) or psychiatric disabilities (*n* = 24, 11.4%) (Table 4).

#### 3.1.2. Comparison between Female Caregivers and Non-Caregivers

No statistically significant differences were detected between the two groups (caregivers and non-caregivers) with regard to socio-demographic variables, such as age, nationality, marital status, having children, level of education, working conditions, and socio-economic status (Table 4).

Caregivers were characterized by a statistically significant higher proportion of loss of loved ones due to COVID-19 compared to non-caregivers (chi-square: 5.396; d.f. 1; *p* = 0.020) (Table 5). No statistically significant differences were revealed between the two groups (caregivers and non-caregivers) with regard to other variables related to life events and clinical characteristics (Table 5). Working caregivers did not complain about the complexity of managing their lives any more than non-working caregivers.

No statistically significant differences were found between the two groups of women with regard to PHQ-9 scores and family functioning dimensions (Table 6).

Compared to non-caregivers, women who had a caregiving role obtained lower scores for six out of the eight health-related quality-of-life domains (with a pre-eminent impact on physical health), including GH (*t*-test: 3.370; *p* = 0.002), PF (*t*-test: 2.556; *p* = 0.011), RF (*t*-test: 2.412; *p* = 0.017), BP (*t*-test: 2.464; *p* = 0.015), MH (t-test: 2.887; *p* = 0.004), and SF (*t*-test: 2.720; *p* = 0.007), while RE (*t*-test: 1.922; *p* = 0.056) and VT (*t*-test: 1.949; *p* = 0.053) scores approached statistical significance (Figure 2).

We subsequently characterized the subsample of caregivers by specifying whether they cared for people with physical (*n* = 23; mean age, years = 39.0 SD = 19.4) or mental disabilities (*n* = 24; mean age, years = 35.3 SD = 18.3), and found that six out of the eight SF-36 dimensions were statistically significant (Figure 3).

An ANOVA test and post hoc analysis provided evidence of the differences between the mean of non-caregivers compared to the two groups of caregivers for GH (ANOVA: F = 5.668; *p* = 0.004), highlighting a greater impairment in women caring for people with physical (Bonferroni correction: mean difference = 11.66278, *p* = 0.028) and psychiatric disabilities (Bonferroni correction: mean difference = 10.59756, *p* = 0.048) (Figure 3).

In the PF dimension, we found evidence of the differences between the mean of non-caregivers compared to caregivers of people with psychiatric disabilities ANOVA: F = 3.676; *p* = 0.027), with the latter displaying poorer physical function (Bonferroni correction: mean difference = 10.68089; *p* = 0.037). Moreover, the RF dimension underlined a marked difference between the mean of female non-caregivers compared to those who cared for a person with psychiatric disabilities (ANOVA: F = 4.761; *p* = 0.001), highlighting the presence of a role limitation related to physical problems amongst caregivers of people with psychiatric issues (Bonferroni correction: mean difference = 21.570; *p* = 0.007) (Figure 3).

With regard to MH, evidence of a significant difference between the mean of women caring for a person with psychiatric disabilities versus non-caregivers was revealed (ANOVA: F = 5.829; *p* = 0.003), highlighting the poorer mental health of those caring for someone with a psychiatric disability (Bonferroni correction: mean difference = 17.50000; *p* = 0.003) (Figure 3).

An analysis of the SF-36 dimensions of VT and SF revealed a significant difference only amongst caregivers of people with mental disabilities versus those caring for a person with physical disabilities and non-caregivers (VT ANOVA: F = 3.488; *p* = 0.003; Bonferroni correction: mean difference = 14.27846; *p* = 0.027; SF ANOVA F = 5.147; *p* = 0.007; Bonferroni correction: mean difference = 17.84807; *p* = 0.005), thus demonstrating how these two specific dimensions were only impaired in this subpopulation (Figure 3).

No statistically significant differences in family functioning were found in the three dimensions identified by the FFQ between the two groups of caregivers (Table 7).

Caregivers of psychiatric patients complained of a higher subjective burden (*t*-test for independent samples: *t* = −3.461, *p* = 0.001) and less support from relatives and friends (*t*-test for independent samples: *t* = −2.256 *p* = 0.029) compared to caregivers of patients with an organic illness (Table 7).

### 3.2. Correlations between Age, Years of Education, Health-Related Quality of Life, Family Functioning, and Burden of Care

Table 8 shows the correlations between the variables of health-related quality of life, family functioning, and burden of care with age and level of education. Age was positively and statistically significantly correlated with GH, MH, RE, VT, and SF, as measured using the SF-36, suggesting that getting older enables greater adaptation, better perception of general health conditions, and improved social functioning. In the same way, level of education was positively and significantly correlated with all quality-of-life dimensions as measured using the SF-36. It was, however, negatively and significantly associated with depressive symptoms, suggesting a protective role for a higher level of education on mental health and quality of life.

Depressive symptoms, as measured using the PHQ 9, seemed to increase over time in caregivers with a lower level of education. Despite their satisfying functional adaptation and positive restructuring of difficulties, they displayed a persistently depressed mood. The correlation analyses revealed significant negative correlations between depressive symptoms and GH, MH, RE, VT, and SF.

Statistically significant negative correlations were found between objective and subjective burden of care, as measured using the FPQ, and perceived quality of life and its related dimensions (GH, MH, ER, VT, and SF), thereby confirming the strong impact of a caregiving role on the participants’ health-related quality of life. However, our data showed that objective burden of care tended to impair the perception of mental health conditions more than subjective burden. The correlation analyses showed negative and significant correlations between a lack of support from friends and MH. Indeed, a lack of support from both friends and relatives and a lack of professional help negatively and significantly correlated with VT, whilst they positively and significantly correlated with depressive symptoms (PHQ 9), confirming the crucial role of social support on the mental health of patients and their caregivers.

With regard to family functioning, good problem-solving and communication skills and achieving personal goals in the family context were positively and significantly associated with age, level of education and all quality-of-life dimensions. In contrast, they were associated negatively and significantly with depressive symptoms (PHQ-9). In addition, these skills seemed to increase over time, along with a lower subjective burden of care and better perception of social and professional support (with the exception of problem-solving skills). Communication skills, however, seemed to increase in line with a better perception of both objective and subjective burden of care. These results further confirm the potential of good problem-solving and communication skills and achievement of personal goals acting as promoters of improved mental well-being, better quality of life, and social /professional support for family members and their caregivers.

### 3.3. Variables Impacting Severe Depressive Symptomatology

Table 9 illustrates the results of the multinominal logistic regression analysis with absent/mild depression symptoms (PHQ-9 = 1, score 0–5) as the dependent variable.

The first set of coefficients comparing women who obtained a score of 1 on the PHQ-9 (absent/mild depression symptoms) and those scoring 2 (range 6–10, moderate depression) revealed two statistically significant predictors. Firstly, women who had lived through the highly traumatic 2009 L’Aquila earthquake were almost four times more likely to suffer from moderate depression. Secondly, older age appeared to exert a protective effect against manifestations of moderate depression when compared to a mild presentation.

Compared to the first set of coefficients, in the second set representing a comparison between women scoring 1 (PHQ-9 range 0–15, absent/mild depression symptoms) and women scoring 3 (PHQ-9 range 11–15, moderately severe depression), the highly traumatic experience of the 2009 L’Aquila earthquake displayed a 9-fold increased predictive probability of onset of moderately severe depression when compared to women displaying mild depressive symptoms. In the sample investigated, previous access to mental health services resulted in a significant 7-fold increase in the likelihood of manifesting moderately severe depression when compared to women with mild depression. Life management difficulties perceived during the COVID-19 pandemic led to an almost 3-fold statistically significant increase in the probability of being affected by moderately severe depression. The protective role exerted by older age against manifesting a more severe depressive psychopathology was confirmed, and an adjunctive variable represented by problem-solving skills in the family context was also identified as a protective factor in this second set of coefficients.

In the third set of coefficients illustrating the comparison between women scoring 1 (PHQ-9 range 0–15, absent/mild depression symptoms) and women scoring 4 (PHQ-9 > 15 severe depression), the more robust variable was previous access to mental health services, which significantly increased the likelihood of exhibiting severe rather than mild depressive symptomatology by 10-fold. A low level of education was identified as a risk variable, accounting for a more than 5-fold probability of manifesting severe depression. Perceived difficulties in life and household management during the COVID-19 pandemic were confirmed as accounting for more than 3-fold statistically significant increase in having severe depression.

In women, older age and good problem-solving skills were confirmed as exerting a predictive protective role against severe depression compared to the manifestation of mild symptoms. In this third set of coefficients, the pursuit of individual goals reached statistical significance as a predictive protective factor in maintaining improved mood and not manifesting severe depressive symptoms.

## 4. Discussion

The current study aimed (1) to evaluate depressive symptomatology, quality of life, and family functioning, with a particular focus on the difficulties faced by women during the COVID-19 pandemic, in a general female population sample by comparing (a) working and non-working women and (b) caregiver vs. non-caregiver women, and (2) to identify factors influencing the onset of severe depressive symptomatology under the framework of a biopsychosocial model approach.

Previous access to mental health services, low level of education, and difficulties in life management during the COVID-19 pandemic were confirmed as variables capable of influencing the onset of severe depressive psychopathology when compared to having a normal mood tone in our sample recruited from the general female population. Older age, good problem-solving skills, and the ability to pursue personal goals were identified as protective factors. Having previously experienced natural catastrophic collective traumas (2009 L’Aquila earthquake) displayed the most robust predictive value related to the presentation of a clinical profile characterized by moderately severe depressive symptomatology, thus suggesting a role of past life events in psychopathological frailty alongside the vulnerability variable of having stressful pre-existing mental health conditions.

In our sample recruited from the general female population, half of the women complained about the complexities of managing their lives during the COVID-19 pandemic when compared to their previous routines. Surprisingly, working women, commonly assumed as being more distressed due to their work–family balance [36], manifested less depressive symptoms, a better quality of life, and more competent problem-solving and communication skills in the family context, alongside an understandable poorer ability to pursue their personal goals, than non-working women.

Compared to European working women, in whom a higher prevalence of depressive symptoms was seen in women working from home compared to those who commuted to their place of work [40], our findings seem to suggest a positive value of the role of “paid work”. Indeed, in our study, stay-at-home women who carried out no-paid work displayed more depressive symptoms and worse health-related quality of life, despite having better functioning in pursuing their personal goals. Limited social contact seemed to be a key factor involved in the presentation of higher levels of depression, with home working resulting in a marked reduction in face-to-face social contacts with family, friends, and colleagues [39], whilst women who continued to work from the office were still able to network to some extent with people outside their household during lockdowns.

With regard to difficulties encountered in life and household management, the findings of our study only partially align with the national survey conducted by the ONDA Foundation in April 2021 in Italy [79]. The comparison is hampered by the diversity of the characteristics of the samples recruited. The national ONDA online survey included 609 women (over 55% from central southern Italy, age range 25–55 years) who, prior to the pandemic, had been in paid employment, many of whom with a university degree (40%), and had been in a stable affective relationship (almost 70%). In our study, although the level of higher education was comparable, more than half of the women were living alone, without having a stable relationship, and around three quarters were not in paid employment or financially independent. Moreover, in the present study, just over 15% of women reported struggling financially, whilst in the ONDA national survey, 39% of working women reported having experienced significant economic challenges following the COVID-19 outbreak, which had mainly affected workers who were contractually less protected, or women who had lost/changed jobs, had their working hours reduced, or had been furloughed, particularly those living in central southern Italy [79].

When investigating mental health conditions, the ONDA 2021 survey reported how, since the start of the pandemic, 85% of women had been affected by at least one mental disorder over a prolonged period and had resorted to treatment. Our study, however, was aimed mainly at assessing the presence of depressive symptoms, thus leading to the detection of a lower 40% rate of other mental issues in our sample based on the PHQ-9 cut-off score of 10 used to identify the presence of moderately severe/severe depression. The data obtained in our study were similar to those reported by Arpino and Pasqualini in Italy, who reported that 47% of the sample evaluated felt depressed during the first COVID-19 lockdown [80].

Our study found that during the COVID-19 pandemic, when compared to non-caregivers, women in charge of the care of a family member with a physical or psychiatric disability complained of worse health-related quality of life, as also reported in a German study [42]. Caregivers of patients affected by mental disabilities showed a statistically significant impairment in vitality and social functioning compared to the other two subgroups of women studied. They complained of a higher subjective burden and less support from relatives and friends than caregivers of patients affected by organic diseases. With regard to objective burden of care, a lack of help from relatives, friends, and professionals was associated with lower family functioning. Our findings align with previous studies highlighting the stressful role of caregivers during the COVID-19 pandemic [45,47,49,50,67,81,82]. Our findings are also consistent with those of recent studies demonstrating how female caregivers experienced mental health issues during the pandemic [81,83], with a multicentric Italian study reporting even higher values than those obtained in a study on caregivers of schizophrenic family members prior to the pandemic [66]. Family distress was higher in households caring for a psychiatric patient, and the increasingly onerous burden for mental health caregivers compared to those with family members affected by physical disabilities confirmed the findings of both Fusar-Poli et al. [47,50] and previous literature data [84]. A limitation of our study related to the assessment of caregivers looking after people with intellectual disabilities, who seemed to experience more significant difficulties during COVID-19 lockdowns than those who cared for people with a mental illness [81].

Within our conceptual framework based on a psychosocial vulnerability model, the deepening of depressive symptomatology during the COVID-19 pandemic was carefully investigated. In the sample investigated, we recorded 20% of women reporting severe depressive symptoms (PHQ-9 > 15), thus prompting our interest in exploring factors that might have contributed to the severity levels.

Compared to women displaying a normal mood tone, the likelihood of manifesting moderate depression was approximately four times higher in women who were exposed to the 2009 L’Aquila earthquake and reported it as a highly traumatic collective experience. This variable increased its predictive power by more than double when women reported moderately severe depression, highlighting a vulnerability to life-event stressors. Our results confirmed the finding that prior traumas seem to increase reactivity to and potential harm of new traumas [71].

Pre-existing mental health conditions increased more than 10-fold the likelihood of exhibiting severe depression, confirming this factor as a risk factor, which is in line with previous studies [71,74,85]. The finding concerning the predictive value of lower educational attainment, which led to a more than 5-fold increase in the likelihood of experiencing severe depression, provides further confirmation of previous literature data [74,85]. The identified risk factor for the manifestation of severe depression, represented by the variable “complex life management during the COVID-19 pandemic”, seemed to include numerous issues related not only to women’s household duties and responsibilities and their “work–life” balance [36], but also to the lack of job and financial independence, the absence of an affective relationship, family lifestyle, and social isolation [75]. Older age, higher education, and better family functioning were identified as protective factors against severe depression and were associated with a reduced burden of care in caregivers. Problem-solving skills, pursuing personal goals, and avoiding overinvolvement in family or other problems were identified as the life skills needed to cope more effectively with the consequence of a deepening depression caused by pandemic-related difficulties. This finding does not yet seem to have been addressed in the literature.

Our initial hypothesis relating to variables capable of influencing severe depression in women was only partially confirmed by our estimated comprehensive psychosocial model. Pre-existing mental health conditions suggest an underlying vulnerability, which heavily influences the manifestation of severe depression. With regard to the role of traumas, our study did not take into consideration exposure to adverse childhood experiences (ACEs) known to act as pervasive risk factors for developing major mental and somatic disorders across the lifespan and for reduced longevity [86]. Nevertheless, the findings obtained from our study population suggest the impact of previous traumas, such as the collective traumatic experience of an earthquake, which seem less enduring and stable than ACEs, but nonetheless play a relevant “retraumatization” role and manifest as the source of psychopathological moderate depression.

### Strengths and Limitations

To the best of our knowledge, this is the first Italian study to evaluate depressive symptomatology during the COVID-19 pandemic in the general female population, taking into account women’s working roles, their experience of caring for disabled people, and the influence of previous catastrophic events.

Furthermore, this study represents a step toward verifying the need for identifying and preventing an escalation of mental health problems due to the pandemic, which is in line with the Sustainable Development Goals proposed by the United Nations Organization, namely Goals 3 and 5.

According to the United Nations, Goal 3 aspires to ensure health and well-being for all. Campaigns and events, such as women’s health events, which are generally promoted by ONDA and which we took part in and reported in this study, contribute to the early identification and destigmatization of mental disorders from a gender perspective. Our Goal 3 is closely related to Goal 5, which includes pursuing gender equality and empowerment, identifying multiple areas of commitment toward addressing women’s challenges, and topical issues for gender equality, including work–life balance and disabled caregiving, which strongly impact women’s lives, not only during a pandemic.

Nevertheless, this study presents several limitations. Firstly, the present study shows a primary limitation in sample recruitment. Street interviews are one of the fastest and most accurate forms of real-time data collection. In this form of data collection, respondents are approached on the street and asked targeted questions. Researchers register their responses using an electronic device such as a tablet, an iPad, or pen and paper. We estimate a potential bias in agreeing to the interview, suggesting that women with problematic symptoms would be more likely to accept the invitation to share their emotional distress. In addition, we estimate the potential influence of social desirability bias, leading to over-reporting of socially desirable behaviors or attitudes, and under-reporting of socially undesirable behaviors or attitudes.

Secondly, we were only able to involve a limited sample of women because of the limited time of the interview.

Thirdly, depression categorizations were based on a questionnaire, the PHQ-9. Although the PHQ-9 is “*an instrument for making criteria-based diagnoses of depressive and other mental disorders commonly encountered in primary care … reliable and valid measure of depression severity*” [60], this type of measure cannot replace a more comprehensive clinical assessment.

Fourth, due to the selectivity of our sample, our findings are also of limited generalizability in view of the territory and the consequences for residents of the catastrophic 2009 L’Aquila earthquake. The evacuation of the town, displacement, temporary accommodation, with more than 10,000 people still living in temporary housing, slow rebuilding [58], the social and economic consequences affecting the community, and the COVID-19 pandemic have led to a situation where, for residents, life in the town has never truly “returned to normal”, as is the case of geographical areas experiencing “unique circumstances and challenges” [71].

Moreover, we did not consider other types of traumatic events that are unfortunately common in women’s lives, such as experiences related to discrimination and/or structural, familial, or partner violence. These events likely increased during isolation and the pandemic [32,87], which could have reduced women’s ability to manage stress and other adverse conditions, affecting their psychological health.

## 5. Conclusions

The present study, conducted in a nonclinical sample of women, investigated the impact of the COVID-19 pandemic on working activities, caregiving responsibilities, and family functioning.

The results obtained suggest that the pandemic produced a more serious impact on the mental health of non-working women and caregivers of family members with pre-existing psychiatric conditions. Therefore, based on our findings, during the pandemic, difficulties in life management did not seem to be intended “tout-court” as a “work–life balance” distress but rather as a more comprehensive distress (“*life is hard*”), presumably due to the lack of job and financial independence, the responsibilities of taking care of relatives, the absence of an affective relationship, and social isolation.

A series of factors concurred to influence the onset of depression in women and the severity of symptoms in response to the pandemic, almost as though the pandemic had partly reflected the cumulative effects of traumas. Interestingly, the value of women’s protective skills against depression, such as problem-solving abilities, pursuing individual goals, and taking care of themselves first without necessarily prioritizing the need of family and others, came to the forefront. These characteristics are an antithesis to those commonly used to describe the stereotypical Italian “woman and mom”.

The findings of our study should be expanded to address further in-depth studies focusing on the complexities of psychopathological frailties in women and their individual and psychosocial strengths after experiencing life stressors and “immersive” traumatic events, such as the COVID-19 pandemic. The ultimate aim will be to translate the findings obtained into evidence-based, gender-specific, personalized clinical and psychosocial strategies for the purpose of improving the mental health of women in everyday life according to the gender medicine approach.

## Figures and Tables

**Figure 1 behavsci-13-00892-f001:**
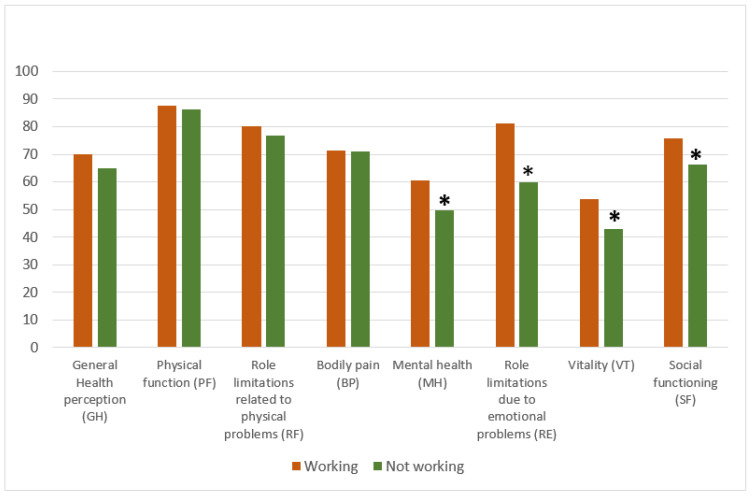
Comparison of health-related quality of life in its eight dimensions in female workers and non-workers. *t*-test, * *p* < 0.05.

**Figure 2 behavsci-13-00892-f002:**
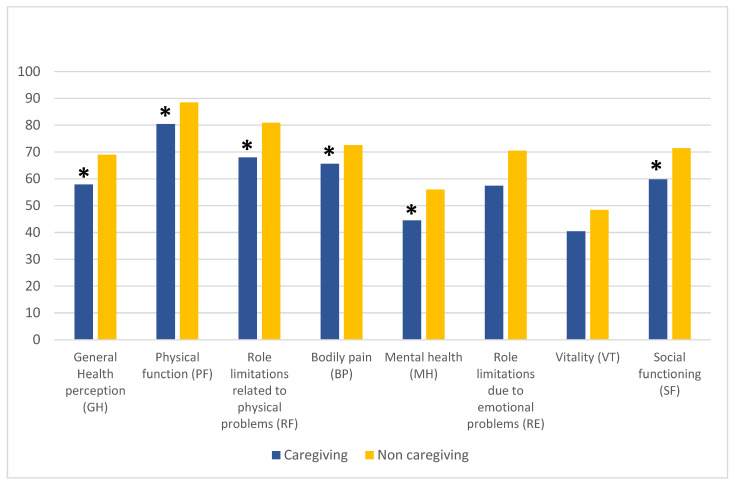
Health-related quality of life in its eight dimensions amongst caregiving and non-caregiving women. *t*-test, * *p* < 0.05.

**Figure 3 behavsci-13-00892-f003:**
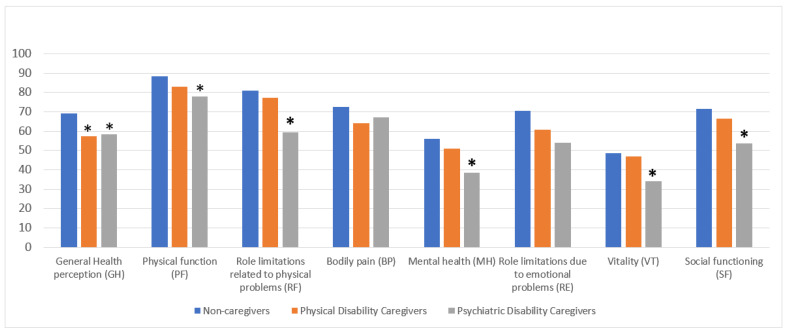
Health-related quality of life in its eight dimensions amongst the three groups of women. ANOVA test, * *p* = < 0.05.

**Table 1 behavsci-13-00892-t001:** Comparison of the socio-demographic and clinical characteristics of the sample (*n* = 211) between working and non-working women (*n* = 211).

Variables Included	Workers (*n* = 73)	Non-Workers (*n* = 138)
Age, mean (SD) *	41.8 (13.2)	32.4 (20.1)
Working conditions (%)		
Self-employed/freelancers	23 (31.5)	
Full-time work	35 (47.9)	
Part-time work	15 (20.5)	
Student	-	91 (65.9)
Housewife	-	13 (9.4)
Unemployed	-	12 (8.7)
Retired	-	22 (15.9)
Nationality (%)		
Non-EU citizens	4 (5.5)	4 (2.9)
Marital status (%) *		
Single	16 (21.9)	77 (55.8)
Married/partnership	49 (67.1)	50 (36.2)
Separated/divorced	7 (9.6)	4 (2.9)
Widowed	1 (1.4)	7 (5.1)
Parents of children (%) *	33 (45.2)	32 (23.2)
Level of education (%) *		
>13 years (graduated)	24 (32.9)	96 (69.6)
Socio-economic status (%)		
High–upper middle income	39 (53.4)	56 (40.6)
Middle–low income	27 (40.2)	57 (41.3)
Struggling financially	7 (9.6)	25 (18.1)

* *p* < 0.05.

**Table 2 behavsci-13-00892-t002:** Comparison of life events and clinical characteristics of the sample of female workers and non-workers (*n* = 211).

Variables	Workers (*n* = 73)	Non-Workers (*n* = 138)
Complex management of family life and work during COVID-19 pandemic (%)	31 (42.5)	77 (55.8)
Infection with COVID-19 (%)	9 (12.3)	12 (8.7)
Refusal of COVID-19 vaccination (%)	6 (8.2)	10 (7.2)
Loss of someone close due to COVID-19 (%) *	7 (9.6)	21 (15.2)
Subjected to the 2009 L’Aquila earthquake (%) (*n* = 100) *	45 (61.6)	55 (39.9)
Loss of someone close during the 2009 L’Aquila earthquake (%) (*n* = 100)	5 (11.1)	10 (18.2)
Severe impact of 2009 L’Aquila earthquake (%) (*n* = 100 women exposed) (intensity: severe; very severe)		
Family life	21 (46.7)	27 (49.1)
Work	18 (40)	13 (23.6)
Social life	21 (46.7)	22 (40)
Severe impairment due to the L’Aquila 2009 earthquake in two out of the three domains investigated (%) (*n* = 43) *	22 (30.1)	21 (15.2)
Previous contact with services due to mental health problems (%) (*n* = 94)	38 (52.1)	56 (40.6)
Mental health problems reported (%)		
Anxiety	20 (27.4)	40 (29)
Family and interpersonal problems	13 (17.8)	26 (18.8)
Depression	14 (19.2)	23 (16.7)
Sleep disorders	9 (12.3)	15 (10.9)
Eating disorders	10 (13.7)	12 (8.7)
Substance abuse	--	3 (2.2)
Other problems	5 (6.8)	14 (10.1)
Treatments		
Admission to a psychiatric ward	--	2 (3.5)
Psychopharmacological treatment (*n* = 39)	11	28
Type of drug		
Anxiolytic drugs	4 (36.4)	8 (28.6)
Antidepressant drugs	6 (54.5)	14 (50)
Antipsychotic drugs	1 (9.1)	6 (21.4)

* *p* < 0.05.

**Table 3 behavsci-13-00892-t003:** Depressive symptomatology as measured using the PHQ-9 and family functioning as measured using the FFQ in the two groups of female workers and non-workers included in the sample.

Variables	Total Sample (*n* = 211)	Workers (*n* = 73)	Non-Workers (*n* = 138)
PHQ-9 total mean score (SD) *	9.5 (6.17)	7.82 (5.7)	10.30 (6.2)
PHQ-9 total score > 10 (%) *	82 (40.2)	19 (26)	63 (45.7)
PHQ-9 score 1–5—absent–mild depression (%)	64 (30.3)	28 (38.4)	36 (26.1)
PHQ-9 score 6–10—moderate depression (%)	65 (30.8)	26 (35.6)	39 (28.3)
PHQ-9 score 11–15—moderately severe depression (%)	44 (20.9)	9 (20.5)	35 (25.4)
PHQ-9 score > 15—severe depression (%)	38 (18)	10 (26.3)	28 (20.3)
Family Functioning Questionnaire (SD)			
Communication *	23.3 (4.8)	24.9 (4.7)	22.5 (4.6)
Problem solving *	21.0 (6.7)	23.6 (6.0)	19.7 (6.7)
Personal goals *	23.8 (3.9)	22.3 (3.8)	24.5 (3.8)

* *p* < 0.05.

**Table 4 behavsci-13-00892-t004:** A comparison of socio-demographic and clinical characteristics in the sample of female non-caregivers and caregivers (*n* = 211).

Variables Included	Non-Caregivers (*n* = 164)	Caregivers (*n* = 47)
Age, mean (SD)	35.3 (18.5)	37.0 (18.8)
Range age		
Young adults (18–35 years) (%)	100 (61)	26 (55.3)
Adults (%)	51 (31.1)	18 (38.3)
Over 65 (%)	13 (7.9)	3 (6.4)
Nationality (%)		
Non-EU citizens	7 (4.3)	1 (2.1)
Marital status (%)		
Single	72 (43.9)	21 (44.7)
Married/partnership	76 (46.3)	23 (48.9)
Separated/ divorced	8 (4.9)	3 (6.4)
Widowed	8 (4.9)	--
Parents of children (%)		
No children	118 (71.3)	29 (61.7)
1 child	15 (9.2)	8 (17.0)
2 children	23 (14.1)	6 (12.7)
3 children	8 (4.2)	4 (10.6)
Level of education (%)		
>13 years (graduated)	70 (42.3)	21 (44.7)
Working conditions (%)		
Self-employed/freelancers	15 (9.1)	8 (17.0)
Full-time work	32 (19.5)	3 (6.4)
Part-time work	11 (6.7)	4 (8.5)
Student	72 (43.9)	19 (40.4)
Housewife	9 (5.5)	4 (8.5)
Unemployed	9 (5.5)	3 (6.4)
Retired	16 (9.8)	6 (12.8)
Socio-economic status (%)		
High–upper middle income	75 (45.7)	20 (42.6)
Middle–low income	66 (40.2)	18 (38.3)
Struggling financially	23 (14.0)	9 (19.1)

**Table 5 behavsci-13-00892-t005:** Life events and clinical characteristics of the comparative sample of female non-caregivers and caregivers (*n* = 211).

Variables	Non-Caregivers (*n* = 164)	Caregivers (*n* = 47)
Complex management of family life and work during COVID-19 pandemic (%)	86 (52.4)	22 (46.8)
COVID-19 infection (%)	16 (9.8)	5 (10.6)
Refusal of COVID-19 vaccination (%)	11 (6.7)	5 (10.6)
Loss of someone close due to COVID-19 (%) *	17 (10.4)	11 (23.4)
Subjected to 2009 L’Aquila earthquake (%) (*n* = 100)	77 (47)	23 (48.9)
Loss of someone close during the 2009 L’Aquila earthquake (%)	12 (15.5)	3 (13)
Severe impact of 2009 L’Aquila earthquake (%) (*n* = 100 women exposed) (intensity: severe; very severe)		
Family life	37 (48)	11 (47.8)
Work	25 (32.4)	6 (26.1)
Social life	32 (41.5)	11 (47.8)
Severe impairment due to the 2009 L’Aquila earthquake in two out of the three domains investigated (%) (*n* = 43)	33 (42.9)	10 (43.5)
Previous contact with services due to mental health problems (%) (*n* = 94)	71 (43.3)	23 (48.9)
Mental health problems reported (%)		
Anxiety	42 (25.6)	18 (38.3)
Family and interpersonal problems	28 (17.1)	11 (23.4)
Depression	27 (16.5)	10 (21.3)
Sleep disorders	16 (9.8)	8 (17)
Eating disorders	17 (10.4)	5 (10.6)
Substance abuse	1 (0.6)	2 (4.3)
Other problems	16 (9.8)	3 (6.4)
Treatments		
Admission to a psychiatric ward	1	1
Integrated treatment (drug prescription + psychotherapy)	14 (8.5)	7 (14.8)
Psychopharmacological treatment (*n* = 39)	29	10
Type of drug		
Anxiolytic drugs	10 (34.5)	2 (20)
Antidepressant drugs	16 (55.2)	4 (40)
Antipsychotic drugs	3 (10.3)	4 (40)

* *p* < 0.05.

**Table 6 behavsci-13-00892-t006:** Depressive symptomatology as measured using the PHQ-9 and family functioning as measured using the FFQ in the two groups of women included in the sample.

Variables	Non-Caregivers (*n* = 164)	Caregivers (*n* = 47)
PHQ-9 total mean score (SD)	9.07 (5.8)	11 (7.1)
PHQ-9 total score > 10 (%)	59 (37.1)	23 (51.1)
PHQ-9 score 1–5—absent–mild depression (%)	51 (31.1)	13 (27.7)
PHQ-9 score 6–10—moderate depression (%)	54 (32.9)	11 (23.4)
PHQ-9 score 11–15—moderately severe depression	35 (21.3)	9 (19.1)
PHQ-9 score > 15—severe depression (%)	24 (14.6)	14 (29.8)
Family Functioning Questionnaire (SD)		
Communication	23.5 (4.4)	22.7 (5.7)
Problem solving	21.4 (6.6)	19.8 (7.1)
Personal goals	24.0 (3.8)	23.0 (4.3)

**Table 7 behavsci-13-00892-t007:** Family functioning as measured using the FFQ and family burden of care as measured using the FPQ in the two groups of female caregivers included in the sample.

	Physical Disability Caregivers (*n* = 23)	Mental Disability Caregivers (*n* = 24)
Family Functioning		
Communication	22.6 (6.5)	22.8 (5.1)
Problem solving	20.6 (4.3)	19.0 (6.5)
Personal goals	22.9 (4.6)	23.0 (4.1)
Burden of care		
Objective burden	1.79 (0.50)	1.86 (0.41)
Subjective burden *	2.00 (0.54)	2.64 (0.70)
Support received from professionals	2.43 (0.92)	2.51 (0.68)
Support received from relatives and friends *	2.24 (0.98)	2.80 (0.69)

* *p* < 0.05.

**Table 8 behavsci-13-00892-t008:** Correlations between age, years of education, and the 5 dimensions of the SF-36 (*n* = 211 women), PHQ-9 total score (*n* = 211 women), the 4 dimensions of burden of care measured using the FPQ (*n* = 47 women), and the 3 dimensions of family functioning (*n* = 211) measured using the FFQ.

Measures	Age	1	2	3	4	5	6	7	8	9	10	11	12	13
1. Education, years	0.304 **	--												
2. SF-36 GH	0.245 **	0.143 *	--											
3. SF-36 MH	0.413 **	0.202 **	0.512 **	--										
4. SF-36 RE	0.380 **	0.322 **	0.451 **	0.628 **	--									
5. SF-36 VT	0.419 **	0.204 **	0.590 **	0.835 **	0.646 **	--								
6. SF-36 SF	0.412 **	0.214 **	0.542 **	0.755 **	0.611 **	0.697 **	--							
7. PHQ9 total score	0.452 **	−0.310 **	−0.560 **	−0.836 **	−0.681 **	−0.799 **	−0.738 **	--						
8. FPQ, objective burden of care	−0.171	−0.0151	−0.384 **	−0.207	−0.354 *	−0.337 *	−0.300 *	0.332 *	--					
9. FPQ, subjective burden of care	−0.076	−0.011	−0.292 *	−0.545 **	−0.391 **	−0.631 **	−0.550 **	0.501 **	0.563 **	--				
10. FPQ, (lack of) support from relatives and friends	0.020	−0.179	−0.207	−0.342 *	−0.155	−0.412 **	−0.231	0.303 *	0.232	0.304 *	--			
11. FPQ, (lack of) professional support	−0.031	−0.186	−0.236	−0.200	−0.139	−0.336 *	−0.199	0.301 *	0.007	0.155	0.580 **	--		
12. FFQ, problem solving	0.434 **	0.291 **	0.422 **	0.519 **	0.399 *	0.556 **	0.529 **	−0.569 **	−0.130	−0.110	−0.465 **	−0.453 **	--	
13. FFQ, communication	0.382 **	0.214 **	0.404 **	0.442 **	0.324 **	0.460 **	0.433 **	−0.472 **	−0.290 *	−0.291 *	−0.373 **	−0.309 *	0.733 **	--
14. FFQ, personal Goals	−0.147 *	−0.167 *	0.350 **	0.208 **	0.131	0.233 **	0.277 **	−0.209 **	−0.261	−0.300 *	−0.403 **	−0.315 *	0.171 *	0.165 *

** *p* < 0.001, * *p* < 0.05 (2-tailed).

**Table 9 behavsci-13-00892-t009:** Logistic multinominal regression with depression as dependent variable. PHQ-9 = 1 (absent/mild depression, score 0–5).

Variables	Categories	B	*p*	Exp(B)	95% Confidence Interval for Exp(B)	
					Lower Bound	Upper Bound
Age	Moderate depression	−0.036	0.010	0.964	0.938	0.991
	Moderately severe depression	−0.095	0.000	0.909	0.868	0.952
	Severe depression	−0.068	0.008	0.934	0.888	0.983
Lack of a stable romantic partnership	Moderate depression	0.124	0.784	1132	0.465	2.756
	Moderately severe depression	1.101	0.053	3.006	0.988	9.148
	Severe depression	1.113	0.098	3.044	0.813	11.399
Less than 13 years of education	Moderate depression	0.0362	0.411	1.436	0.605	3.408
	Moderately severe depression	0.306	0.591	1.358	0.444	4.152
	Severe depression	1.688	0.021	5.410	1.288	22.714
Struggling financially	Moderate depression	−0.427	0.579	0.652	0.144	2.952
	Moderately severe depression	0.870	0.260	2.387	0.525	10.852
	Severe depression	1.224	0.145	3.402	0.654	17.687
Previous access to mental health services	Moderate depression	0.634	0.155	1.885	0.787	4.514
	Moderately severe depression	1.956	0.001	7.070	2.288	21.842
	Severe depression	2.391	0.000	10.923	2.908	41.020
Traumatic experience with the 2009 L’Aquila earthquake	Moderate depression	1.369	0.013	3.932	1.332	11.612
	Moderately severe depression	2.242	0.004	9.416	2.077	42.691
	Severe depression	1.705	0.077	5.502	0.829	36.527
COVID-19 infection	Moderate depression	0.239	0.742	1.270	0.307	5.248
	Moderately severe depression	−0.073	0.939	0.930	0.143	6.039
	Severe depression	−0.483	0.651	0.617	0.076	5.008
Complex life management during the COVID-19 pandemic	Moderate depression	0.533	0.221	1.705	0.726	4.003
	Moderately severe depression	1.086	0.046	2.964	1.018	8.628
	Severe depression	1.280	0.045	3.598	1.026	12.616
Caregiving for a loved one	Moderate depression	−0.129	0.821	0.879	0.287	2.693
	Moderately severe depression	0.304	0.652	1.355	0.362	5.075
	Severe depression	1.058	0.150	2.880	0.683	12.147
Problem solving	Moderate depression	−0.093	0.096	0.912	0.817	1.017
	Moderately severe depression	−0.164	0.010	0.849	0.750	0.961
	Severe depression	−0.234	0.001	0.791	0.686	0.912
Communication	Moderate depression	−0.095	0.192	0.909	0.788	1.049
	Moderately severe depression	−0.029	0.726	0.971	0.824	1.145
	Severe depression	−0.124	0.189	0.883	0.734	1.063
Personal goals	Moderate depression	−0.100	0.100.	0.905	0.803	1.019
	Moderately severe depression	−0.127	0.105	0.881	0.756	1.027
	Severe depression	−0.222	0.012	0.801	0.674	0.952

The reference category is 1. PHQ-9 total score for absent/mild depression: 0–5; moderate depression = 2 (PHQ-9 scores 6–10); moderately severe depression = 3 (PHQ-9 scores 11–15); and severe depression = 4 (PHQ-9 scores > 15). Statistically significant values are in bold.

## Data Availability

The data presented in this study are available from the corresponding author upon request.

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
