# Peer review of "Deepening Depression in Women Balancing Work–Life Responsibilities and Caregiving during the COVID-19 Pandemic: Findings from Gender-Specific Face-to-Face Street Interviews Conducted in Italy"

_behavsci, 2023, doi:10.3390/bs13110892_

Round 1

Reviewer 1 Report

Comments and Suggestions for Authors

In the manuscript, the authors reported a study on mental health and daily life stress among women during the COVID-19 pandemic. The study was carefully conducted and the findings were well presented, which have a great amount of practical implications. However, there are a few issues that should be addressed (see below for detailed comments).

1. In the introduction section, the association between loneliness and its adverse psychological outcomes should be elaborated more within certain theoretical frameworks. Especially, the pandemic have long-lasting psychological effects that is multi-faceted. It may be better to begin with the focus on mental health (the forth paragraph), and move loneliness (the second paragraph) to a later section.

2. The purpose of the section "2.1. Context" is not clear. Usually, the Method section should start with the description of the sample.

3. Please clarify the type of sampling. 

4. Was there any information about race/ethnicity of the sample?

5. Low levels of daily life functioning seems to be an important contributing variable to depressive symptoms. Would the author be able to provide the information about possible intervention to modify the variable in order to improve women's mental health?

Comments on the Quality of English Language

The English expression is fine, and only moderate editing is needed.

Reviewer 2 Report

Comments and Suggestions for Authors

The reserachers have done a good work in introducing a new variable to the discussion, that of the effects of the Aquila's Earthquake on the well being of women studied. 

Previous access to mental health services, a lower level of education, and difficulties in managing life during the COVID-19 pandemic were confirmed Previous access to mental health services, a lower level of education, and difficulties in managing life during the COVID-19 pandemic were confirmed as factors that could influence the development of severe depressive psychopathology compared to individuals exhibiting a normal mood tone in our sample derived from the general female population.

Strengths of the research:

Identification of specific risk factors (e.g., limited access to mental health services, low education) contributing to severe depressive psychopathology during the COVID-19 pandemic.

  1. Recognition of protective factors (e.g., older age, strong problem-solving skills, pursuit of personal goals) that can mitigate the risk of severe depressive symptoms.
  2. Highlighting the significance of prior exposure to collective traumas (e.g., the 2009 L’Aquila earthquake) as a robust predictor of moderately severe depressive symptomatology, emphasizing the role of past life events in psychopathological vulnerability alongside pre-existing mental health conditions.

Comments on the Quality of English Language

Strengths of the research:

  1. Identification of specific risk factors (e.g., limited access to mental health services, low education) contributing to severe depressive psychopathology during the COVID-19 pandemic.
  2. Recognition of protective factors (e.g., older age, strong problem-solving skills, pursuit of personal goals) that can mitigate the risk of severe depressive symptoms.
  3. Highlighting the significance of prior exposure to collective traumas (e.g., the 2009 L’Aquila earthquake) as a robust predictor of moderately severe depressive symptomatology, emphasizing the role of past life events in psychopathological vulnerability alongside pre-existing mental health conditions.

Weaknesses of the research:

  1. The study focuses exclusively on a sample of females from the general population, potentially limiting the generalizability of findings to other demographic groups.
  2. The research does not delve into the specific mechanisms by which these variables contribute to depressive psychopathology, leaving questions about causality unanswered.
  3. While the study identifies protective factors, it may not provide comprehensive strategies or interventions to address the risk factors for severe depressive psychopathology during pandemics.

Author Response

Thank you for the observations reported for the strengths and weaknesses identified. Regarding weaknesses of the research, in particular, the lack of analysis of specific mechanisms by which examined variables contribute to depressive psychopathology, the objective was to identify the factors that contribute more than others to the onset of depressive symptoms in a general population of women, without analyzing mediating factors or, in any case, underlying mechanisms, which can be investigated through other analyses. We underlined the importance of a gender medicine-based approach in the conclusions section.

Reviewer 3 Report

Comments and Suggestions for Authors

Reviewer 4 Report

Comments and Suggestions for Authors

First, I would like to thank the authors for the opportunity to read this paper. The subject is interesting and engaging. The issue of depression In Women Balancing Work-Life And Caregiving During The Covid-19 Pandemic is both important and contemporary in every country.

However, I would like to make a few suggestions for the authors to improve the present quality of the paper. Thus, a double check regarding the sources listed in the References part would be recommended, in order to verify if all sources are cited in the content of the article. Also, a distinct Literature review part could be included, beside the Introduction section.

The conclusion part should be extended.

Author Response

We verified the references cited in the article. The five simultaneous revisions of this work (somewhat non-homogeneous) do not allow for further emphasis on the topics covered in the text.

Reviewer 5 Report

Comments and Suggestions for Authors

The paper provides insightful evidence about the impact of the COVID pandemic on mental health, quality of life, and family functioning in a sample of the general female population. Anyway, there are some issues related to the unclear research gap and the coverage of literature review that still need some improvement. I encourage the authors to consider the comments given below and revise the paper accordingly in order to enhance the overall quality and completeness of the paper.

(1)   The research gap and contributions of the study need to be emphasized in the introduction. 

(2)   Some typos are detected in the article. The paper needs to be carefully proofread.

(3)   The background information in the introduction about the psychological impacts from COVID-19 pandemic need to be elaborated in more detail. The authors should provide some examples of psychological problems that people experience from the pandemic by considering the following papers as the references.

a. How Does Mindfulness Help University Employees Cope with Emotional Exhaustion during the COVID-19 Crisis? The Mediating Role of Psychological Hardiness and the Moderating Effect of Workload, Scandinavian Journal of Psychology. 63(5), 449-461. https://doi.org/10.1111/sjop.12826

b. Stress and coping during COVID-19 pandemic: Result of an online survey. Psychiatry Research, 295, 113598. https://doi.org/https://doi.org/10.1016/j.psychres.2020.113598

c.  Teachers, Stress, and the COVID-19 Pandemic: A Qualitative Analysis. School Mental Health, 15(1), 78-89. https://doi.org/10.1007/s12310-022-09533-2

d. Effects of Workplace Rumors and Organizational Formalization During the COVID-19 Pandemic: A Case Study of Universities in the Philippines, Corporate Communications: an International Journal, 26(4), 793-812. https://doi.org/10.1108/CCIJ-09-2020-01

e.  Stress and anxiety among university students in France during Covid-19 mandatory confinement. Comprehensive Psychiatry, 102, 152191. https://doi.org/https://doi.org/10.1016/j.comppsych.2020.152191

Author Response

  • 1) The five simultaneous revisions of this work (somewhat non-homogeneous) do not allow for further emphasis on the topics covered in the introduction.
  • 2) Thank you for your suggestion. The identified typos were corrected.
  • 3) Thank you for your suggested references. We selected and integrated it into the text.
  • Please see the attachment,

Round 2

Reviewer 3 Report

Comments and Suggestions for Authors

The authors have diligently addressed the majority of the comments and suggestions made during the initial round of revision, resulting in a notably improved version of the manuscript.

I can only recommend, if convenient, revisiting the format of the tables in the manuscript. The current presentation retains some SPSS-specific aspects, and it would be beneficial to refine the table formatting for consistency with the overall document style and to ensure clarity and visual appeal.

Furthermore, I advise the authors to consider for further studies conducting an analysis of normality and performing a sample size calculation. For future studies, assessing the normality of the data would yield valuable insights into the distribution of the sample, potentially influencing the choice of statistical methods and the interpretation of results. Additionally, calculating an appropriate sample size is crucial for ensuring that the study possesses adequate statistical power to detect meaningful effects, thus reinforcing the reliability of the findings.

In summary, the manuscript have improved in clarity and organisation. These enhancements significantly contribute to the overall quality of the research. 

Reviewer 5 Report

Comments and Suggestions for Authors

Overall, the authors have made the satisfactory revision. However, the authors still ignored some recommended papers that were previously suggested to be included in the review about the the psychological impacts from COVID-19 pandemic. 

- How Does Mindfulness Help University Employees Cope with Emotional Exhaustion during the COVID-19 Crisis? The Mediating Role of Psychological Hardiness and the Moderating Effect of Workload, Scandinavian Journal of Psychology. 63(5), 449-461. https://doi.org/10.1111/sjop.12826
